# Evaluation of Oak-Specific Consumption, Efficiency, and Losses from an Aesthetic Veneer Factory

**Gneorghe Cosmin Spirchez [1], Aurel Lunguleasa [1,*] and Valentina Doina Ciobanu [2]**

[1] Department of Wood Processing and Design of Wood Products, Faculty of Wood Engineering, Transilvania University of Brasov, 1 Universitatii Street, 500068 Brasov, Romania; cosmin.spirchez@unitbv.ro
[2] Department of Silviculture and Forest Engineering, Transilvania University of Brasov, 1 Sirul Beethoven Street, 500123 Brasov, Romania; ciobanudv@unitbv.ro
*   Correspondence: lunga@unitbv.ro

**Featured Application: The results obtained in this paper can be applied practically in any aesthetic veneer factory, in this sense with the working methodology and also the values of the losses, efficiency, and specific consumption obtained in the case of oak logs being important.**

**Abstract:** The paper aimed to investigate the losses on the manufacturing flow of an aesthetic veneer factory, in order to know their value and to take measures to limit or use them efficiently. For the analysis, the oak species (*Quercus robur*) was taken into account, through 25 analyzed logs. Statistical investigation has used the Minitab 18 program, for a 95% confidence interval. Minimum values of losses were obtained for the sectioning-grooming operations with 0.6% and debarking with 4.9%, and the highest values were obtained when cutting veneers with 15.9% and formatting veneers with 22.8%. Based on the losses on the manufacturing flow of 67.3%, the specific consumption index of 2.75 $m^3/m^3$ or 1.84 $m^3/1000\ m^2$ of veneer was determined when the average thickness of the veneers was 0.67 mm. The paper highlights the methodology and the values resulting from the investigation of the technological losses and of the specific consumption from an aesthetic veneer factory.

**Keywords:** aesthetic veneer; loss; efficiency; oak; specific consumption

## 1. Introduction

In any log processing factory (be it of timber, chipboard, fiberboards, technical or aesthetic veneer, etc.) it is necessary to know the manufacturing losses for each technological operation, but also for the total factory, the main goal being to reduce losses and maximize the amount of primary product. In the alternative, an additional purpose of knowing these losses is to use these losses judiciously, finding the simplest and most economical method of use. In direct correlation with these technological losses, two other global characteristics of the factory are defined, such as the yield of the raw material use (efficiency) and the specific consumption index, defined with the following two calculation relations:

$$E = 100 - \sum L_i\ [\%] \tag{1}$$

$$I_{sc} = 100 \div E\ [m^3\ logs/m^3 veneers] \tag{2}$$

where E is efficiency, in %; $L_i$ is losses for each technological operation, in %; and $I_{sc}$ is index of specific consumption, in $m^3/m^3$.

In order to obtain superior efficiency in aesthetic veneers, it is necessary to know in detail the technological operations in the factory, so to use all methods of reducing losses and obtaining maximum quantities of aesthetic veneers. In the scientific literature, the way of expressing the losses in percentages and the efficiency also in percentages is used. For example, if one knows the value of the total losses of 52% at the factory level, one can immediately find an efficiency of 48% and can show that from a quantity of 100 $m^3$ logs

one will obtain the method of percentage losses $0.48 \times 100 = 48$ m$^3$ veneer and a specific consumption index of $1/0.48 = 2.082$ m$^3$ logs/m$^3$ veneer.

Specific to this technology of aesthetic veneers, but especially due to the fact that their price is given in per unit area, is that the specific consumption can be expressed in per unit area of veneer, respectively, per 1000 m$^2$ of veneer. For this expression, a block of 1 m$^3$ of veneer sliced into several veneers of a certain thickness is considered. It turns out that the number of veneers can be easily obtained from this 1-m$^3$ block of veneer, dividing the height of 1000 mm of the block by the thickness of the veneer. Based on these considerations, it can be determined that 1 m$^3$ of veneer could have the following surface:

- 1000 m$^2$, when veneer thickness is 1 mm,
- 1428 m$^2$, when veneer thickness is 0.7 mm,
- 2000 m$^2$, when veneer thickness is 0.5 mm,
- 3333 m$^2$ when veneer thickness is 0.3 mm.

If one considers the above specific consumer index (2.082 m$^3$ logs/m$^3$ veneer), one will obtain an index of the specific consumption per unit area as follows:

- 2.083 m$^3$ logs/1000 m$^2$ veneers, when veneer thickness is 1 mm,
- 1.458 m$^3$ logs/1000 m$^2$ veneers, when veneer thickness is 0.7 mm,
- 1.041 m$^3$ logs/1000 m$^2$ veneers, when veneer thickness is 0.5 mm,
- 0.624 m$^3$ logs/1000 m$^2$ veneers, when veneer thickness is 0.3 mm.

These values are given for similar moisture content of logs and veneers; otherwise, the shrinkage of the wood must be taken into account. It is observed that the technology of aesthetic veneers has a series of particularities of expressing the specific consumption, first of all, in relation to the ratio to 1000 m$^2$ of veneer, which depends on the thickness of the veneers (and the thickness of the veneers depends on the wood species; porous and soft wood will have the greater of the veneer thickness). It is also noted that the main solution to reduce specific consumption is to reduce the thickness of aesthetic veneers to 0.2–0.3 mm, an action that is not always possible for all wood species. Based on the above considerations, the following relation of the specific consumption index per unit area can be deduced:

$$I_{sc1} = 100 \times t \div E \text{ [m}^3 \text{ logs/1000 m}^2 \text{ veneers]} \tag{3}$$

where $I_{sc1}$ is specific consumption index per unit area; E is efficiency, in %; and t is thickness of veneers, in mm.

The specific consumption of the raw material in the manufacture of aesthetic veneers is dependent on the wood species, being 2.2 m$^3$/m$^3$ for beech and oak and 1.5 m$^3$/m$^3$ for exotic species, mainly depending on the diameter of the logs, because with increasing the diameter of the logs the losses will decrease. The general balance of the use of raw material (logs) in an aesthetic veneer factory that delivers oaks is: veneer 36%, sawdust 4%, log bark 4%, sides 2%, veneer residue 23%, knife residue 24%, and various other losses 8% [1].

Dumitrascu et al. [2] analyzed the influence of oak log defects on the efficiency of log transformation into aesthetic veneers, finding that the diameter and dormant buds have the greatest influence on efficiency. Bernaczyket al. [3] used densified veneers at temperatures of 20 °C, 120 °C, and 180 °C, at a pressure of 1 MPa, for 2 min to obtain plywood. The use of polyvinyl adhesive contributed to a relatively high shear strength value of 2.2 N/mm$^2$, and the use of PPE adhesive contributed to an increase in shear strength to 2.5 N/mm$^2$. Pramreiter et al. [4] used birch veneer with L = 1000 mm, l = 100 mm, and t = 0.55 mm. It was established that the dynamic modulus of elasticity was on average 14% lower than the static modulus of elasticity. Bekhta et al. [5] highlighted the aesthetic properties of different species of thermo-mechanical densified veneer. These properties were highlighted by gloss measurements. The characteristics of the veneers had an influence on the roughness and gloss. Ahmed et al. [6] pointed out that, in veneer drying plants, controllers facing shortcomings act in regulating the optimal veneer temperature.

Kawlerczyk et al. [7] highlighted how the impregnation of birch veneer with a mixture of potassium carbonate and urea affected the properties of fire plywood, shear strength, modulus of elasticity, and modulus of rupture. The veneers were impregnated with aqueous, flame-retardant solution in proportions of 20% and 30%. Impregnation did not affect the modulus of elasticity or the modulus of rupture. Impregnation caused a decrease in the adhesion of the veneers to the plywood with UF adhesive. Buchelt et al. [8] pointed out that, for the production of bonded veneer composites, they are coated with cellulose or layers of paper. The authors described the mechanical properties of such a composite consisting of beech veneer with thickness of 0.35 mm and a layer of cellulose with a thickness of 0.12 mm, glued with PVAc adhesive. Other authors [9] created a guide for obtaining three-layer veneer laminates with a stable shape. A curvature test was performed on plywood from pine veneer sheets (*Pinus radiata*). Plywood formation temperature and pre-formation moisture content have the greatest influences. Dietzel et al. [10] presented a methodology for determining the properties of elastic material on decorative veneers. Tensile tests were performed at different temperature and moisture content values of the veneers.

Fekiak et al. [11] showed that 3D modelling of veneers, as opposed to modelling of plastic or other materials, is limited due to the characteristics of wood. Water and ammonia-water solutions were used. The veneers were tested by immersion in cold water (20 °C), hot water (95 °C), and a 25% ammonia solution for different times. Experimental results showed that 3D modelling of veneers increased by 66–119% after plasticization with a 25% ammonia solution as opposed to veneers with a moisture content of 7.65%. Franke et al. [12] conducted research on the impregnation of beech veneers with low- and medium-molecular-weight phenol-formaldehyde solutions. Low-molecular-weight phenol-formaldehyde-impregnated veneers showed a higher modelling than high-molecular-weight veneers. Herold and Pfriem [13] showed that, in order to increase the casting performance, the veneer samples were impregnated with furfuryl alcohol for improved plasticization. The degree of plasticization of furfuryl alcohol is comparable to the use of water. Pfriem and Buchelt [14,15] studied the comparison of veneer subjected to mechanical tests, respectively, parallel and perpendicular strengths on fibers. The modulus of elasticity perpendicular to the fibers was smaller than that parallel to the fibres.

Dupleix et al. [16] presented the importance of optimal heating temperatures by soaking beech, birch, and spruce. It is necessary to heat the green wood before peeling to improve the peeling process, as well as the quality of the veneer. Zerbst et al. [17] highlighted the importance of tensile and shear tests for veneers under normal conditions but also immersed in water. Kallakas et al. [18] investigated the effects of disposing of different black alder and poplar veneers by replacing birch veneer in plywood with cores of alternately growing species. The results showed that alder and birch plywood have the highest bending strength. The birch veneer had an adhesive consumption of 152 g/m², while the trembling poplar veneer had an adhesive consumption of 179 g/m². The low-density wood veneer had a higher consumption of adhesive. Reichel et al. [19] presented the importance of used veneers and cutting speed, but also the importance of veneer preconditioning inside of veneer technology. Gerencser and Bejo [20] presented the importance of water jet cutting as a possible alternative to traditional cutting methods. The width of the veneer decreased as the water jet propagated into the material. Fang et al. [21] presented the modification of the flexibility of the decorative veneer with the help of plastic foil. This method is effective because it widens the application area and the strength of the veneer decreased when its moisture content was higher than 20%.

Pramreiter et al. [22] presented the importance of the development of veneer-based wood composites due to the high flexibility of design and mechanical strength compared to solid wood. The study showed the influence of veneer thickness on tensile strength compared to solid wood. Ropes with a thickness of 0.5 mm, 1 mm, and 1.5 mm and thin timber with a thickness of 1.5 mm, 3 mm, and 5 mm were used. From the experimental results it was concluded that the tensile strength was 70% higher than veneer lumber.

Salca et al. [23] presented the properties of plywood made of densified and un-densified veneer. The plywood was made of black alder and birch veneer, using urea formaldehyde (UF). MOE increased with increasing plywood density. The increase in veneer density led to a decrease in MOR. The temperature of 150 °C for the densification of the veneer was sufficient to obtain improved plywood properties. Svoboda et al. [24] presented the characteristics of beech and poplar wood plywood, with thicknesses of 6 mm, 10 mm, and 18 mm, when vinyl acetate was used for gluing. The study concluded that the use of carbon fibre and fiberglass led to increased plywood performance.

Menezzi et al. [25] presented the evaluation of the potential use of the wood species *Scizolobium parahyba* in the production of laminated veneer lumber LVL. The non-destructive wave test method was used to determine the EdV (dynamic veneer modulus). The authors noted that the higher the EdV values, the higher the mechanical strength values of the LVL. Bekhta et al. [26] analyzed the effect of veneer densification on the change of temperature inside the plywood during the hot-pressing stage. Birch veneer and phenol-formaldehyde resin were used to make the plywood samples. Plywood structures from densified and non-densified veneer layers, with and without adhesive, were investigated. Plywood made of 3, 5, 7, 9, and 11 layers was made in laboratory conditions. It was found that plywood made of densified veneer will reduce the plywood pressing time by 2–29%, depending on the number of layers of veneer. The shear strength values of the densified veneer plywood were twice higher than the standard value of 1 MPa (EN 314-1).

From the analysis of the bibliographic studies, we deduced a general conclusion, namely, that, although there are consistent laboratory studies in the field of aesthetic veneers, there are very few studies in situ, respectively, inside some aesthetic veneer factories. There is also a total lack of work in the field of specific consumption of raw materials and losses on the manufacturing flow. That is why this work wanted to analyse the manufacturing losses from an aesthetic veneer factory in order to know their values and to calculate the global, specific consumption indicators of the factory. This study helps the factory management to better coordinate its activity of managing the secondary resources in the factory.

## 2. Materials and Methods

Investigations on losses, efficiency, and specific consumption index were performed at the company LOSAN Romania SRL (Brasov, Romania) on batches of 25 pieces of round oak wood (*Quercus robur*) at which losses were quantified at each point in the technological flow, respectively, for the following technological operations: log debarking, grooming and sectioning, shaping of logs, cutting of veneers by flat cutting, drying of veneers, and formatting of veneers. The logs were purchased from Romania and Croatia, respectively, from areas as close as possible to the location of the veneer factory. The diameter at the thin end was 40–55 cm and the average length was 3.4 m, with a variation of 2.7–3.6 m. Of the three quality classes, only class B of logs was taken, with a taper less than 1 cm/m, a curvature less than 15%, and a maximum ovality of 8%.

Logs were sorted and sectioned according to their length into cut-logs corresponding to the opening length of the planer cutting machines. Additionally, the ends of the defective logs were removed and, on the outside, the traces of logs and gallstones were removed. The percentage losses were determined as the ratio between the volume of material that fell during this operation (the volume of losses, in the form of log ends, sawdust, and other pieces) and the volume of logs taken into work, with the following equation:

$$P_i = (V_i - V_f) \div V_f \times 100 \ [\%] \tag{4}$$

where $P_i$ is the value of losses, in %; $V_i$ is initial volume of the log before the technological sectioning operation, in m$^3$; and $V_f$ is the final volume of the hubs, after the sectioning operation, in%.

The debarking of the logs, with the analysis of each log, was done on the debarking ramp with a milling machine, after the thermal plasticization treatment of the logs, and

meant the quantification of the losses in the form of bark or wood waste that diminished the initial volume of the log. For this, 25 logs were taken, finding the volume of wood that was lost from each log. In order to obtain the losses, a percentage ratio was made between the volume of losses from the logs and the initial volume of the log considered, with the Equation (4). The logs were shaped on prisms, in order to obtain larger quantities of tangential veneer. This operation, performed with a horizontal band saw, resulted in losses of longitudinal edges and sawdust. The amount of percentage loss was determined by the same relation. The shaped logs were transformed into veneer, resulting in manufacturing residues in the form of knife residue and undersized veneer remains. The veneers were cut with a vertical plane cutting machine, fixing the prism by vacuum and multi-nail plate (Figure 1).

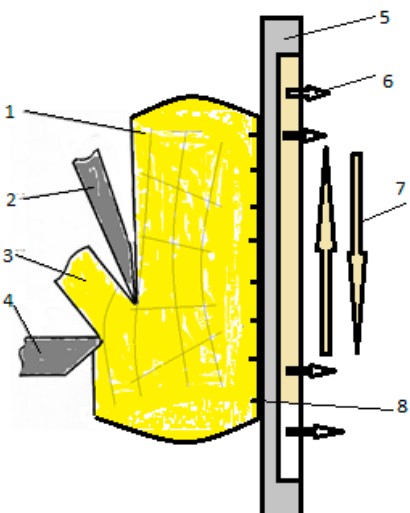

**Figure 1.** Plan—vertical cutting of aesthetic veneers: 1—prism; 2—knife; 3—veneer; 4—press bar; 5—prism mounting plate; 6—vacuum prism fixing; 7—alternative rectilinear vertical movement; 8—zone of multi-nail prism fixing.

The losses by drying the veneers were due to their shrinkage from moisture content of over 30% at the time of introduction into the dryer, up to a moisture content of 8% at the exit of the dryer with belt and nozzles. The volume losses of all veneers obtained from logs were determined, but the reporting was the same as the previous cases to the initial volume of the log from which the veneers were obtained, with the help of a similar relationship with Equation (4).

For the verification of the data obtained experimentally, the coefficient of volumetric shrinkage for oak was used from Fibre Saturation Point up to 8%, obtained on 10 specimens of $100 \times 20 \times 20$ mm, extracted from the wood of this oak species and using the principle expressed in Figure 2 and the following relation:

$$S_{v(USF-8)} = ((V_{max} - V_{min}) \times (FSP - MC_8)) \div (V_{max} \times FSP) \times 100 \ [\%] \tag{5}$$

where $V_{max}$ is the maximum volume of the test piece, before drying, in mm$^3$; $V_{min}$ is the minimum volume of the test piece, after drying in the oven at 105 °C, for 12 h; FSP-MC$_8$ is the moisture content difference between fibre saturation point and veneer moisture content after drying of 8%; and FSP is the fibre saturation point (usually 30%), in%.

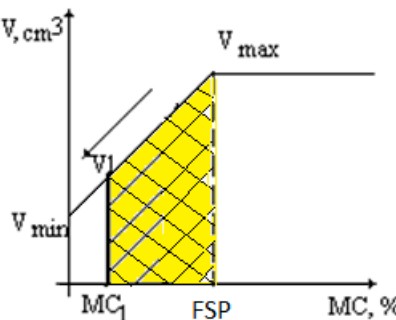

**Figure 2.** Diagram of veneer shrinkage.

The veneers obtained were tangential and had three quality classes depending on the admissibility of the defects and the dimensions of the packages, respectively, 175 × 14 cm for the first class, 120 × 12 cm for the second class, and 60 × 10 cm for the third class. The average thickness of the veneers was 0.67 mm with a standard deviation of ± 0.06 mm. The calculation of the residues from the veneer formatting was done on all the packages obtained from each log. The packages of 12 veneers were formatted in length and width, the remains resulting from the guillotine scissors being in the form of veneer strips. The reporting of the veneer residues [27,28] was done as in the previous cases to the initial volume of each log considered, with a similar relationship as Equation (4).

The analysis of the losses on the manufacturing flow was first performed tabularly, after which it proceeded to the statistical processing of the results. The individual values of the losses obtained at each technological operation were statistically processed, obtaining the arithmetic mean of the values and their standard deviation. Based on the individual losses, the sum of the total losses, the percentage efficiency, and the index of use of the raw material for obtaining aesthetic veneers were obtained.

For the statistical processing of the experimental values, the Minitab 18 program (Coventry, United Kingdom) was used, together with its interpretation graphs such as probability plot, empirical CDF, histogram, and I-MR chart. Through these graphs we obtained as statistical parameters the average survey and standard deviation, but also other parameters of influence, for a 95% confidence interval and, respectively, an error $\alpha = 0.05$.

## 3. Results

Results were expressed graphically, on the base of the registered tabular values.

### 3.1. Losses When Cutting Logs

The Empirical Cumulative Distribution Function (CDF) from Figure 3, performed to determine log section losses, highlights the fit of each experimental value with the cumulative distribution curve. It is observed that "S curve" is asymptotic both at the horizontal line of the value 0% and at the horizontal line of 100%, and the real values approach the empirical curve, thus demonstrating the normal distribution of values.

Based on the average value of the section losses of 0.7%, of the standard deviation of 0.097%, and of taking over two standard deviations for a confidence interval of 95%, the limiting interval of 0.616–0.894% was obtained.

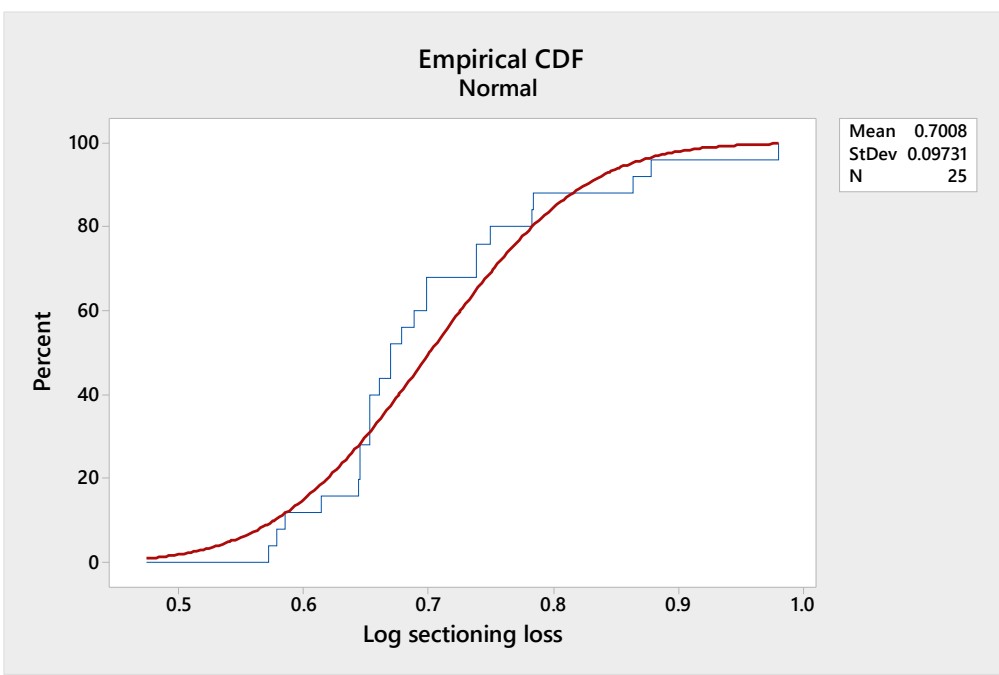

**Figure 3.** Empirical CDF for loss of log sectioning.

### 3.2. Loss on Debarking Logs

The results for this operation with losses, respectively, the debarking operation on the proper ramp, focused on the data collected in the corresponding tables, are visible in Figure 4. An average loss of 4.9% was observed, with a standard deviation of 0.18%, for those 25 lots of logs examined. This value is significant, even if it seems to be very small. The small value of this peeling loss is given by the relatively small thickness of the oak bark (*Quercus robur*) of 4–5 mm and also by the diameter of the logs used, with a range of 52–87 cm [1].

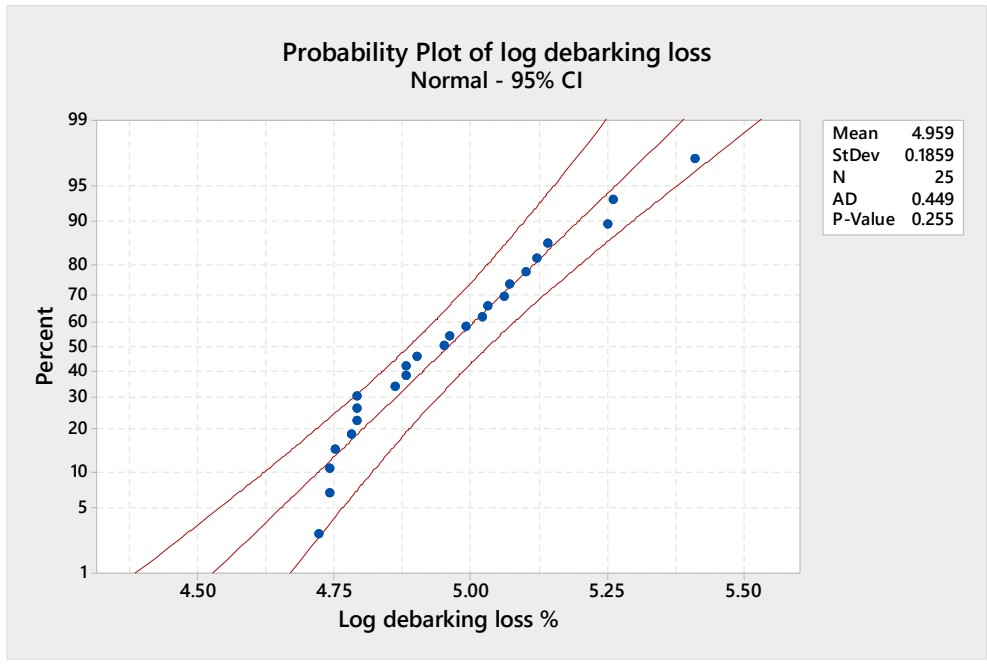

**Figure 4.** Loss when peeling logs.

The graph from Figure 4 was obtained for a 95% confidence interval (respectively, for an error $\alpha = 0.05$) and a normal distribution, observable by framing all values between the two extreme limits. The two parameters of the graph, respectively, the Anderson–Darling (AD) and *p*-value, were interpreted together, because the used the experimental data to determine if null hypothesis was rejected or alternative hypothesis was accepted (respectively, if it falls within a normal distribution). It was observed that the *p*-value of 0.225 was higher than the error value of 0.05, which confirmed the normality of the distribution. By knowing the value of the standard deviation of 0.18 and the confidence interval of 95% (corresponding to two standard deviations), the interval of the values' variation (4.589–5.329)% was also determined. Similar values were obtained by other authors [1,2].

### 3.3. Losses in the Grooming-Sectioning Operation of the Logs

The probability plot diagram for cleaning and sectioning (Figure 5) shows that all experimental values fell within the extreme limits, i.e., the normality of the distribution without aberrant values was confirmed. Moreover, based on the same graph and for a 95% confidence interval, the limits of the lower and upper values of variation of losses for grooming-sectioning were determined, respectively, 0.860–1.203%.

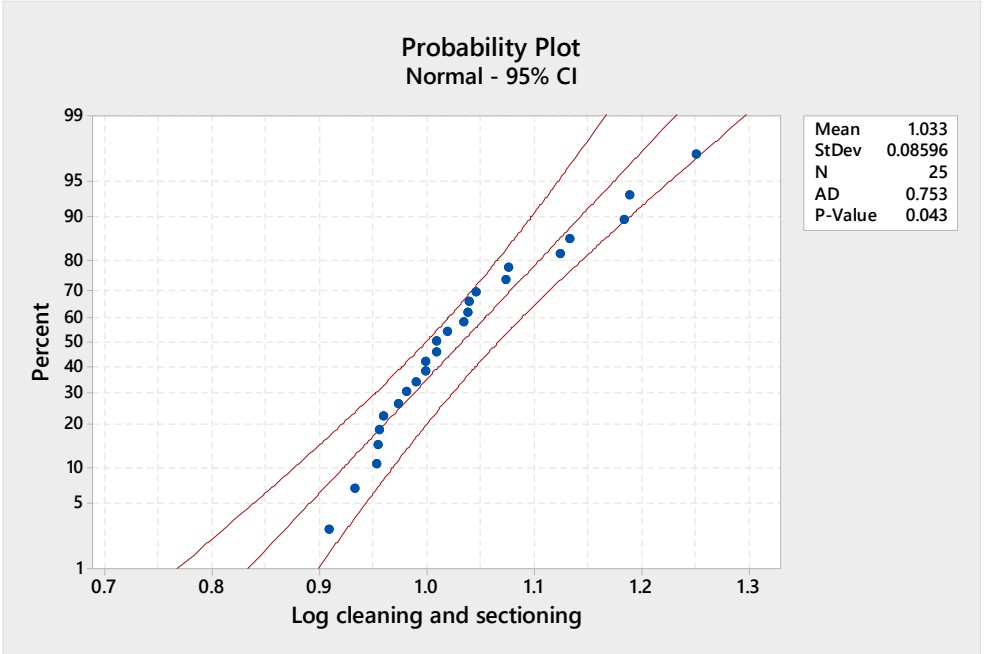

**Figure 5.** Probability plot for log cleaning and sectioning.

### 3.4. Losses from Log Shaping (Trimming)

The Empirical Cumulative Distribution Function (CDF) for log trimming loss in Figure 6 is a curve in the shape of the letter "S" and represents the cumulative percentage proportion of each value in the total values, given that the actual values of losses were found on the Ox axis. To obtain this curve, the first percentage ratio between the first value and the total number of values on the vertical axis were ordered, and the second percentage ratio was added to the first and so on until the last ratio, the percentage at which the value of 100% was obtained. The proximity of the real values to the empirical curve was observed and the extreme values of the losses by shaping were found for 95% confidence interval of 7965–10,347%.

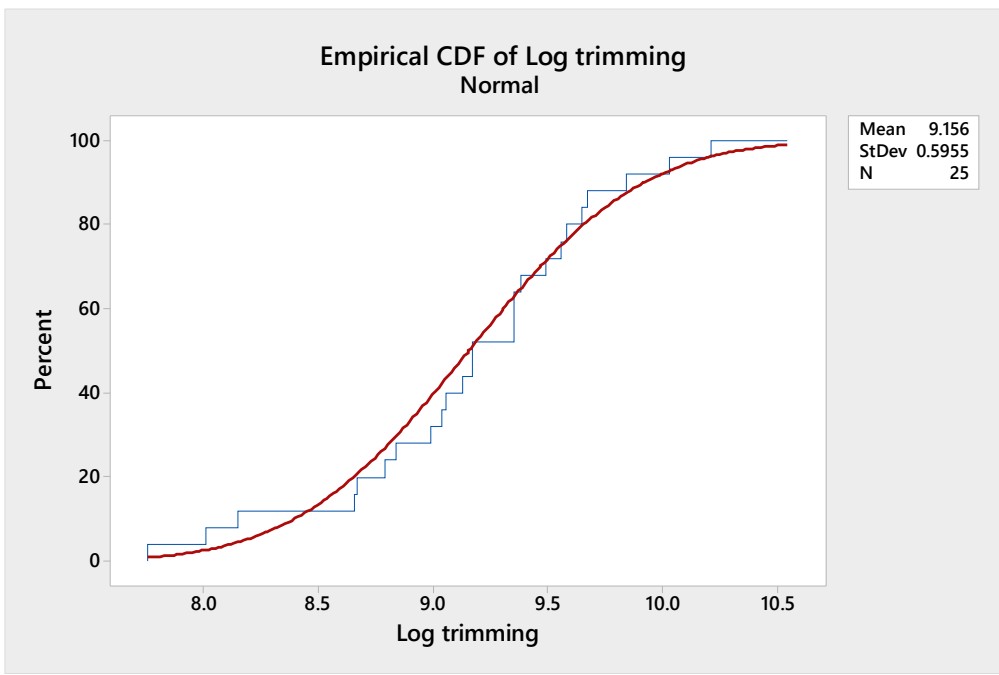

**Figure 6.** Empirical CDF for log trimming loss.

### 3.5. Losses from Veneer Cutting

The framing of the values of the losses in the form of a knife residue from the flat cutting of the veneers between the two control limits (Figure 7) showed the normal distribution of these values. Also, the values of the Anderson–Darling coefficient (AD) and *p*-Value confirmed the normal distribution of values. Based on this diagram, the values of the loss values range of 13.76–18.22% for 95% confidence interval (plus/minus 2 standard deviations) were found. This loss was quite high, being ranked second after the losses from veneer formatting.

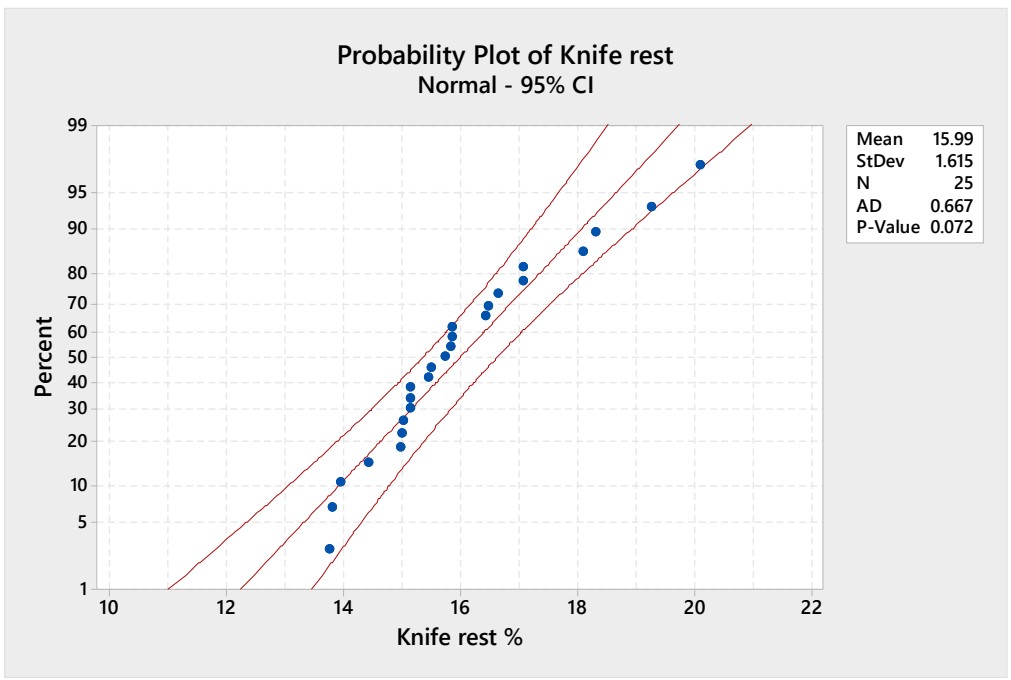

**Figure 7.** Probability plot of knife rest from flat cutting of aesthetic veneers.

### 3.6. Losses from Veneer Drying

The Cumulative Distribution Function (CDF) curve of the drying of the veneers (Figure 8) resulting from the 25 logs examined, by approximating the real points to the empirical values, had a normal distribution of values. Based on the two statistical parameters of trend (arithmetic mean) and spread (standard deviation), as well as by establishing a confidence interval of 95%, a range of variation of the values of drying losses (6.088–9.288%) was obtained.

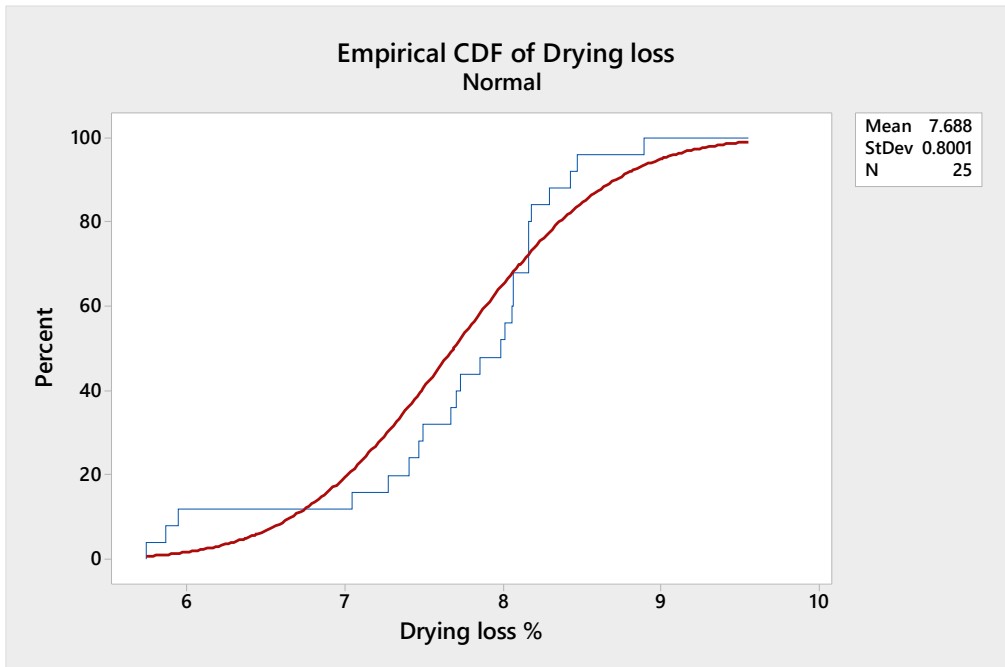

**Figure 8.** Empirical CDF of loss on veneer drying.

Based on the value of the volume shrinkage coefficient on the moisture content range 8–30% that was determined with Equation (5), a value of 7.61% was obtained, very little less than the one obtained experimentally, but which fell within the limits imposed by ±5%.

### 3.7. Losses from Veneer Package Formatting

Figure 9 shows the probability plot curve for expressing losses from veneer formatting. The average value of 22.88% was the largest of the total losses from the manufacture of aesthetic oak veneers. The normal distribution of the values of these losses was observed by framing the values between the two limits on the diagram, but especially by the correlation between the Anderson–Darling coefficient (AD = 0.264), *p*-value of 0.667, and the error $\alpha = 0.05$. Based on the mean value and the standard deviation, the loss interval of (17.87–27.89%) was obtained for a 95% confidence interval.

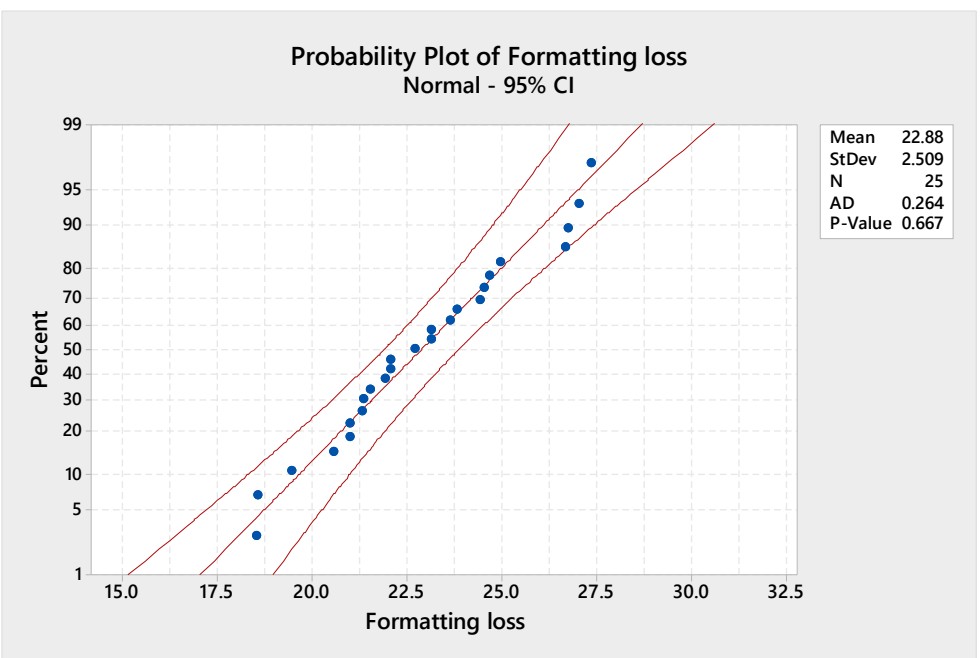

**Figure 9.** Probability plot for veneer formatting loss.

*3.8. Total Losses*

Based on the averages obtained for each technological operation, but also on the individual values of the 25 logs taken into account at each loss, the total technological losses were found by summation in the form of 25 values that were statistically interpreted using the histogram in Figure 10. The normal distribution of values was observed and, based on the average and the standard deviation expressed in this histogram, the variation interval of 60.41–66.99% was found.

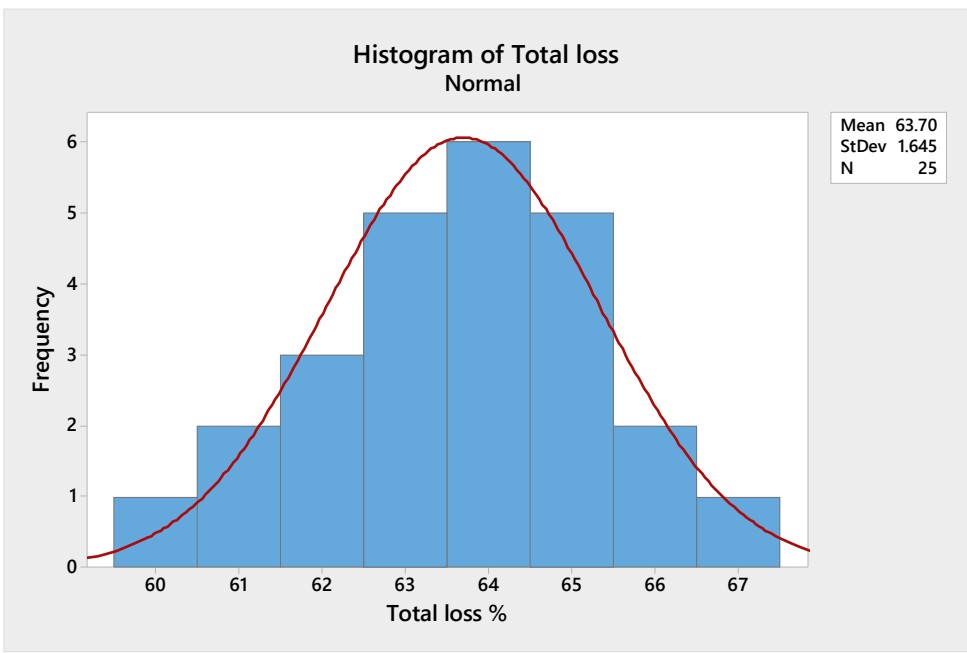

**Figure 10.** Histogram of veneer total loss.

Individual-Moving Range (I-MR) for total loss (Figure 11) is individuated by two diagrams, the first for individual values and the second for moving range. These control diagrams show us that the technological process of making aesthetic veneers was under

control because all the values were arranged between the two control limits, respectively, LCL (low control limit) and UCL (upper control limit). The control limits in this case (both diagrams) were taken for a probability of occurrence of values of approximately 99.97%, respectively, for a range of plus/minus 3 standard deviations. The second diagram was made by moving the first interval so that the lower limit (LCL) was equal to zero. For this, the value of each loss was subtracted from the average of the values (first diagram in Figure 9), thus obtaining another 25 values, closer to zero if the difference was small or further away from zero if the difference was larger. A new average of values of 1.911% and an upper limit of 6.243% were also obtained. These values were confirmed by other researchers [9,14].

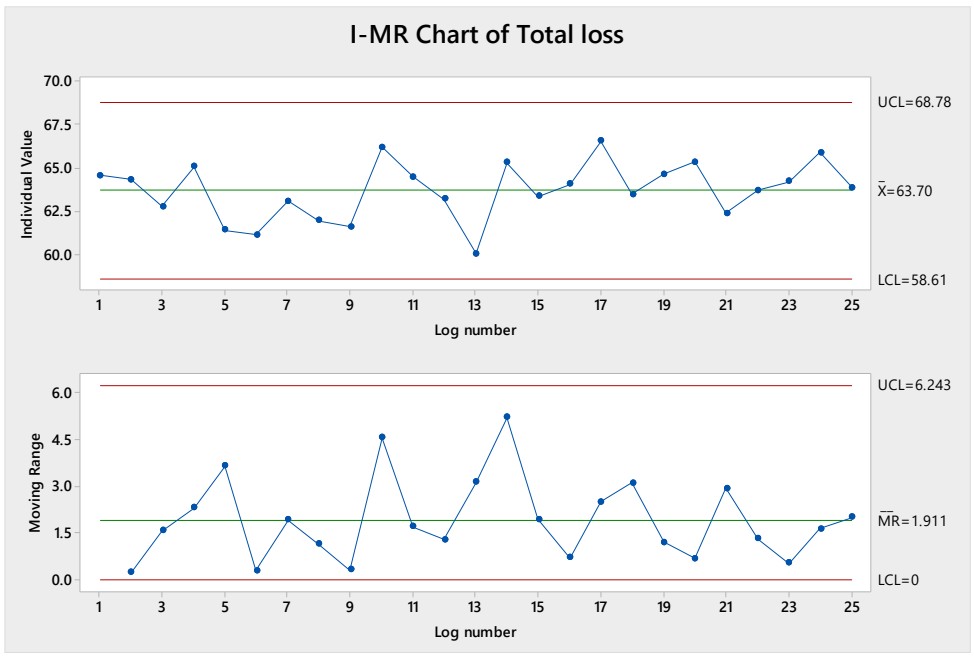

**Figure 11.** I-MR chart for total loss.

Following the in situ analysis of the manufacturing losses in the aesthetic oak veneer factory, a total loss of 63.7% was obtained. Based on this, a percentage efficiency of 36.3% was determined with the help of Equation (1) and, with the help of Equation (2), a specific consumption index of 2.754 $m^3$ logs/$m^3$ veneers was also obtained. If we take into account the average thickness of oak veneers of 0.67 mm at the delivery moisture content of 8%, a specific consumption index per unit area was obtained of 1.845 $m^3$ logs/1000 $m^2$ veneers. Based on the data from this research, respectively, from the losses obtained in the form of veneer, the management staff of the veneer factory designed and dimensioned a pelletizing line of the veneer residues, thus increasing the turnover of the factory by about 15%.

## 4. Conclusions

Through methodology, results, and statistical interpretation, the paper highlights the determination of the values of manufacturing losses in order to find ways of superior recovery and re-use. The efficiency values for the batch of 25 oak logs studied were between 1% for the sectioning operation and 22.8% for the veneer formatting operation. A total loss of 63.7% was obtained, as well as a specific consumption index of 1.84 $m^3$ log/1000 $m^2$ veneer or 2.754 $m^3$ logs/$m^3$ veneers. The resulting values of the losses obtained on the manufacturing flow of the aesthetic oak veneers and based on Minitab statistical program helped the factory management to keep them under control and/or to find methods of their superior capitalization, and also to be able to correctly organize the activity of supplying the factory with logs.

**Author Contributions:** Conceptualization, A.L. and G.C.S.; methodology, G.C.S.; software, A.L.; formal analysis, V.D.C.; investigation, V.D.C.; writing—original draft preparation, A.L.; writing—review and editing, A.L.; visualization, A.L.; supervision, V.D.C. and G.C.S. All authors have read and agreed to the published version of the manuscript.

**Funding:** This research received no external funding.

**Institutional Review Board Statement:** Not applicable.

**Informed Consent Statement:** Not applicable.

**Data Availability Statement:** The data presented in this study are available on request from the corresponding author.

**Acknowledgments:** We would like to thank the Transilvania University of Brasov and the Losan Romania SRL Company, especially Eng. Ciprian Morosanu, for all the support provided in conducting the research and drafting the paper.

**Conflicts of Interest:** The authors declare no conflict of interest.

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
