# Peer review of "Evaluation of Oak-Specific Consumption, Efficiency, and Losses from an Aesthetic Veneer Factory"

_applsci, doi:10.3390/app11094300_

Round 1

Reviewer 1 Report

I find this paper very interesting as a professional paper, especially for the manufacturing company used as a research polygon, but I think the sample is too low to be able to make any scientific conclusions. I would encourage authors very much to extend this research to more than one research polygon.

Additionally, the construction of the paper should be changed, for an example – line 27 – 64 from my point of view should be part of the materials and methods section; line 85 – and forward is more appropriate for literature review section or introduction.

Conclusion is to general and scientific contribution of the paper is insufficient.

The English language should be improved.

Author Response

Reviewer 1

  1. Reviewer: I find this paper very interesting as a professional paper, especially for the manufacturing company used as a research polygon, but I think the sample is too low to be able to make any scientific conclusions. I would encourage authors very much to extend this research to more than one research polygon.

Author answer: The minimum number of specimens for adequate statistical processing is 25-30 pieces. That is why we consider that the results obtained in the paper are satisfactory. In the future, the authors will extend this research to more samples and more industrial units for obtaining more results of aesthetic veneer losses and create a new course chapter for the students of the profile faculties, in order to value the research results from this paper and other similar papers.

  1. Reviewer: Additionally, the construction of the paper should be changed, for an example – line 27 – 64 from my point of view should be part of the materials and methods section; line 85 – and forward is more appropriate for literature review section or introduction.

Author answer: Lines 27-64 are introductory elements, which introduce the reader to the topic of paper, even if some concrete data are presented. These data are given as examples and are not related to the results of our research. For a specialist in the field of aesthetic veneer manufacturing are not necessary, but these introductory values are necessary for readers who do not have all the knowledge necessary to understand the research conducted in this paper. Lines 85-181 belong to the Introduction chapter.

  1. Reviewer: Conclusion is to general and scientific contribution of the paper is insufficient.

Author answer: The paragraph with conclusions was completed, so as to become more explicit, less general and more consistent (see new version of revised paper).

  1. Reviewer: The English language should be improved.

Author answer: The English language was improved all over the paper.

Reviewer 2 Report

It's a good job.
Improve conclusions. In the conclusions you present the results. Lines 407-410 can be a conclusion.

Author Response

Reviewer 2

  • Reviewer: It's a good job. Improve conclusions. In the conclusions you present the results. Lines 407-410 can be a conclusion.
  • Author answer: The paragraph with conclusions was completed, so as to become more explicit, (see new version of revised paper). In this place a paragraph was deleted and a new paragraph was introduced, as a general conclusion of the paper.

Reviewer 3 Report

Dear authors

the work presented for review is interesting and deals with current issues related to the quality of wood raw material used. The work itself concerns the evaluation of the efficiency of veneer at different stages of production. The production of veneer concerns one of the least efficient processes in the production of wood-based materials. Although the resulting product obtains a relatively high price, its production is inefficient. The results themselves are clearly presented and well-illustrated in the figures. The authors also correctly evaluate the obtained test results for the individual stages of the production process. The authors use percentage changes of losses in the paper. For the assessment of the conducted research itself, this is sufficient. However, I suggest enriching the paper with:

- a dimensional description of the tested logs,

- brief information whether the quality and dimensions of the processed logs are representative for the processed raw material, if not, what share they represent in the production?

- try to show the production cycle graphically,

- characterize more the final form of the veneer,

In this way the reader will obtain additional information and make more analyses on his own. Plants with similar production may have similar problems, but on this basis it is not clear whether the information contained can help them.

The authors should also reword the abstract a bit. The last sentence should rather not be such a strongly emphasised conclusion but rather an introduction to the need for this type of analysis.

Author Response

Reviewer 3

The work presented for review is interesting and deals with current issues related to the quality of wood raw material used. The work itself concerns the evaluation of the efficiency of veneer at different stages of production. The production of veneer concerns one of the least efficient processes in the production of wood-based materials. Although the resulting product obtains a relatively high price, its production is inefficient. The results themselves are clearly presented and well-illustrated in the figures. The authors also correctly evaluate the obtained test results for the individual stages of the production process. The authors use percentage changes of losses in the paper. For the assessment of the conducted research itself, this is sufficient. However, I suggest enriching the paper with:

  1. Reviewer: a dimensional description of the tested logs,

Author answer: In the Material section of the paper a new paragraph with the characteristics of the logs was introducedThe logs were purchased from Romania and Croatia, respectively from areas as close as possible to the location of the veneer factory. The diameter at the thin end was 40-55 cm, and the average length was 3.4 m, with a variation of 2.7-3.6 m. Of the 3 quality classes, only class B logs were taken, with a taper of less than 1 cm/m, a curvature of less than 15% and a maximum ovality of 8%”.

  1. Reviewer: brief information whether the quality and dimensions of the processed logs are representative for the processed raw material, if not, what share they represent in the production?

Author answer: In order to eliminate the influence of the defects of the logs on the quality of the veneers and implicitly on the manufacturing losses, only logs of class B quality were taken over. See the paragraph in the chapter on methods and materials.

  1. Reviewer: try to show the production cycle graphically,

Author answer: We consider that the production cycle of the factory is very simple (a series of operations), a graphic realization of it would unjustifiably increase the work, without bringing anything new. In fact, on page 187 are presented the main operations of the technological flow in their natural order.

  1. Reviewer: characterize more the final form of the veneer,

Author answer: The next paragraph was introduced at line 243:The veneers obtained were tangentially and had 3 quality classes depending on the admissibility of the defects and the dimensions of the packages, respectively 175x14 cm for the first class, 120x12 for the second class and 60x10 for the third class. The average thickness of the veneers was o.67 mm with a standardized deviation of +/- 0.06 mm.”

  1. Reviewer: In this way the reader will obtain additional information and make more analyses on his own. Plants with similar production may have similar problems, but on this basis, it is not clear whether the information contained can help them.

Author answer: This paper cannot solve all the problems found in aesthetic veneer factories. The work is carried out in conditions specific to the LOSAN factory, cutting only aesthetic oak veneers by flat cutting, and using class B logs. However, we claim that the work should be a start in this field, by method, but especially by statistical processing of results.

  1. Reviewer: The authors should also reword the abstract a bit. The last sentence should rather not be such a strongly emphasised conclusion but rather an introduction to the need for this type of analysis

Author answer: The last sentence in the abstract has been changed.

Round 2

Reviewer 1 Report

A paper is improved, now. It can be published.